# Regulation of E-Cigarettes in the United States and Its Role in a Youth Epidemic

**DOI:** 10.3390/children6030040

**Published:** 2019-03-04

**Authors:** Mark A. Gottlieb

**Affiliations:** Public Health Advocacy Institute at Northeastern University School of Law, Boston, MA 02115, USA; ma.gottlieb@northeastern.edu

**Keywords:** electronic cigarette, addiction, regulation, vaping, nicotine, ENDS, FDA

## Abstract

During the first decade of federal regulation of electronic nicotine delivery systems (ENDS), the e-cigarette industry has rapidly grown. Recently, the U.S. Surgeon General and Commissioner of the Food and Drug Administration each declared the rapid rise in rates of youth using these products to be an “epidemic.” While a foundational basis for regulating ENDS has been in effect since 2016, deferred enforcement has contributed to acute rise in use by youth. The Agency has undertaken several initiatives to address the problem and warned manufacturers that if current youth trends continue, it will be “game over.”

## 1. Introduction

Few pediatricians had ever heard of an “electronic cigarette” when the first attempt to regulate electronic nicotine delivery systems (ENDS) in the United States was undertaken by the U.S. Food and Drug Administration (FDA) in 2008. The concept seemed promising: an alternative to high-risk combusted tobacco products for addicted smokers designed to reduce their exposure to tobacco-specific nitrosamines and other harmful constituents present in tobacco smoke [1]. But by the end of 2018, the use of ENDS by youth had been declared an “epidemic [2].” The increase in teen use of ENDS is of acute concern. The National Institute on Drug Abuse’s Monitoring the Future survey found that rates of 12th grader vaping in the past year jumped from 27.8% in December of 2017 to 37.3% in December of 2018 [3]. Worse, with no clear treatment protocol for treating addicted teens, pediatricians are lacking reliable tools to help their vaping patients [4]. A review of the regulatory efforts aimed at controlling the marketing of ENDS, a highly addictive product, particularly at the federal level, explains much about why we are now facing an epidemic. New regulatory proposals may shape the public health response.

## 2. Review of the Regulatory History of ENDS

In the early 2000s, Chinese pharmacist and inventor Lik Hon patented the “electronic atomization cigarette”, which “only contain(ed) nicotine without harmful tar [5].” The device used a piezoelectric ultrasound element to aerosolize a liquid nicotine solution [6]. By 2008, shipments of ENDS utilizing this basic design were being imported into the United States, although the market was still relatively small. The FDA, which approves new drugs introduced in the market [7], suspected that these products were drugs or drug delivery devices sold without approval in violation of the Food, Drug and Cosmetic Act and subsequently added three Chinese manufacturers to an Import Alert to seize and halt the importation of e-cigarettes [8]. Two distributors of ENDS filed a lawsuit and sought an injunction to permit these products to be imported and sold in the U.S. [9].

The logic of the FDA’s defense was pretty straightforward: these are products intended to affect the structure or function of the body that require approval as new drugs in order to be sold. At the time FDA detained these shipments, the agency did not yet have regulatory authority over tobacco products. That would occur soon enough. In June of 2009, President Obama signed the Family Smoking Prevention and Tobacco Control Act into law [10]. This long-awaited FDA authority over tobacco products provided for a new Center for Tobacco Products at the agency, which would undertake a range of regulatory and enforcement duties. 

The court ruled that Congress intended a broad definition of “tobacco products” under the new law and that, if FDA wanted to regulate ENDS, it would have to do so as a tobacco product, rather than as a drug and drug delivery device [9]. The FDA appealed the decision to the U.S. Court of Appeals for the District of Colombia, but the lower court’s ruling was affirmed [11]. While ENDS do not actually contain any tobacco, the nicotine in these products is derived from tobacco and, consequently, they are now considered “tobacco products” under federal law [12]. Manufacturers seeking to make therapeutic claims for ENDS would still require approval through the same drug regulatory pathway as other nicotine replacement therapy products [11]. 

It was at this point, when the FDA’s appeal failed, that the opportunity to proactively develop a responsible public health policy response to ENDS was derailed. The regulatory challenge that this posed to the FDA was that Congress only granted it regulatory authority over cigarettes, roll-your-own tobacco, and smokeless tobacco products. Products such as ENDS, cigars, or hookah would need to be “deemed” to be tobacco products through a formal rulemaking process in order for the agency to regulate them [10]. The FDA first seized ENDS shipments in 2008 and, by the time the agency declined to appeal the appellate decision, is was 2011. During that short time, global sales of ENDS had increased approximately 10-fold, dramatically increasing the challenge of regulating the fast-growing industry [13]. 

The FDA did not publish a proposed deeming rule to begin formally regulating ENDS until 2014 [14]. The proposed rules, which were adopted in May of 2016 and effective in August of that year, required ENDS to undergo a “premarket” review process for New Tobacco Products as well as requiring a prominent warning on packaging stating that the products contain the addictive chemical nicotine [15]. During the period of time from the court’s decision until rulemaking began, the business and culture around vaping accelerated rapidly into a billion-dollar industry [16]. Also, the large cigarette manufacturers, derisively known collectively as “Big Tobacco,” began to introduce their own products or simply purchase smaller ENDS manufacturers [17]. With thousands of new products already on the market, the genie was very much out of the bottle and not very likely to get back in. 

As a strategy to gain regulatory control of a dynamic and unregulated market, the Deeming Rule’s procedure for premarket review and approval of ENDS was to permit any ENDS already on the market as of August 8, 2016 to continue to be sold and to allow manufacturers three years to complete a pre-market new tobacco product application and allow no new products on the market during this deferred approval period [15]. In 2017, the time allowed for submission of applications was extended from three to six years from the 2016 effective date of the rule [18]. This delay in enforcement has, in effect, allowed thousands of ENDS, including all ENDS used by youth, to remain on the market for years, despite never being reviewed or authorized by FDA. A legal challenge to this extension filed by public health groups and pediatricians is pending [19].

It was in the two years immediately following the effective date of the deeming rule, however, that youth use of ENDS went from being a serious concern to a declared epidemic [2]. One particular ENDS product appeared to spearhead the rapid increase on youth use of ENDS: Juul. This product, which was introduced in 2015, captured approximately 75% of the total U.S. ENDS market by the end of 2018 [20]. There is much discussion and ongoing research seeking to understand exactly how and why Juul Labs’ product, more so than any other, has had such a dramatic impact on the ENDS market and youth prevalence. Its proprietary chemistry and development of “nicotine salts,” which deliver nicotine in a protonated rather than a free-base form, is certainly a part of the explanation [21]. As such, Juul delivers a different sort of nicotine than did its ENDS predecessors. This is achieved through the use of benzoic acid to lower the pH of Juul’s liquid nicotine solution in order to aerosolize the drug in its protonated form of highly palatable ultrafine particles [21]. Its physical design, marketing approach, and timing may help explain its appeal to youth [22]. The extraordinary success of Juul culminated with Altria, owner of Philip Morris and the world’s most popular cigarette, Marlboro, buying a 35% stake in manufacturer Juul Labs for $12.8 billion at the close of 2018 [23]. Concern extends beyond Juul as copycat products have been flooding the market providing lower cost options for teens [24]. Juul Labs has sought to stop this practice by filing complaints with the U.S. Federal Trade Commission as well as litigation alleging patent infringement in over a dozen instances [25].

The FDA has been urged to fast-track its regulatory response to create controls for the ENDS industry [26,27]. At the same time, FDA was doubling the time for ENDS manufacturers to apply for pre-market approval from three to six years, CDC data showed that, while combusted tobacco youth among high school students remained steady 2011–2016, use of ENDS increased rapidly [28]. After the appointment of a new commissioner in 2017, the agency announced its intention to consider reducing the nicotine level in cigarettes to minimal or non-addictive levels while, “encouraging development of innovative tobacco products that may be less dangerous than cigarettes [29]”. In June of 2018, FDA Commissioner Gottlieb said that, “[w]e believe that if more adults are able to fully transition from combustible tobacco products to ENDS, we might be able to significantly reduce the overall morbidity and mortality associated with tobacco use [30]”. While such statements might be considered to be something of an endorsement of the purported harm reduction potential of ENDS, the agency has taken several steps and articulated a range of potential regulatory responses to the youth vaping epidemic. 

## 3. Regulatory Enforcement Actions and Options under Consideration in 2018/2019

In 2018, FDA began a flurry of activity to address rising concerns over youth vaping. It began an aggressive enforcement action against retailers that were found to have sold Juul or other ENDS to minors, resulting in over 1300 warning letters issued to violators in 2018 [31]. It also made a formal request of Juul Labs (and several other ENDS manufacturers) to, “submit documents relating to marketing practices and research on marketing, effects of product design, public health impact, and adverse experiences and complaints related to JUUL products [32]”. This was followed by the extraordinary step of an FDA raid on Juul Labs’ headquarters, including the seizure of digital data held there [33]. 

One of the provisions of the Deeming Rule was to allow ENDS that were already on the market as of the rule’s effective date of August 8, 2016 to remain on the market pending the application and premarket review process [15]. It appeared, however, that new products were being introduced to the market subsequent to the rule’s effective date and incorporated delivery systems similar to Juul’s [24]. In October of 2018, FDA sent letters to 21 ENDS manufacturers as part of an investigation as to whether they were introducing new products to the market in violation of federal law [34]. In a press release announcing the investigation, the agency said it would, “not allow the proliferation of e-cigarettes or other tobacco products potentially being marketed illegally and outside of the agency’s compliance policy, and we will take swift action when companies are skirting the law [35]”. 

FDA also submitted letters to the five companies that supply more than 95% of the U.S. ENDS market, asking them to each provide a plan to the agency within 60 days detailing how they will “address the widespread youth access and use of their products [31]”. If the responses were inadequate, FDA indicated that it might revisit its policy of allowing these products into the market without an approval order [31]. Shortly after the FDA letters were sent, Altria, maker of the MarkTen e-cigarette product line, announced that it would stop selling most flavored ENDS entirely [36]. It was only two months after that announcement, however, that it purchased its $12 billion stake in Juul Lab [23]. 

Just after the 60 days for responses passed, Commissioner Gottlieb announced that he was directing the agency to develop a policy to restrict all flavored ENDS, other than mint, menthol, and tobacco flavors, to age-restricted retail locations or online only with heightened age-verification requirements [37]. At the time of this writing, no details regarding the timing and manner of implementation of this policy has been released nor is it clear what sorts of heightened age-verification requirements would be successful. Such a policy would presumably remove most flavored ENDS from convenience stores, gas stations, and other common retail settings.

2019 began with a stark warning from the FDA commissioner to the ENDS industry. In a public hearing on January 18, Gottlieb stated that if youth vaping rates continue to rise in 2019, “[i]t will be game over for these products until they can successfully traverse the regulatory process [38]”. The implication is that FDA will remove some or all of these ENDS products from the market pending approval of premarket review applications. Such an action would be a great blow to the ENDS industry and a boon to children’s health. 

There is some question as to whether ENDS products will be authorized for sale by FDA as New Tobacco Products. Rather than the usual “safe and effective” standard that the FDA applies, tobacco products are evaluated with a broader public health standard. It requires FDA to evaluate the risks and benefits of the product to the population as a whole [39]. Considering the recent onset of a teen vaping epidemic combined with the lack of clear treatment strategies for youth addiction, it may well be that manufactures cannot surmount the public health standard that FDA authorization of Premarket Tobacco Applications would require. Such an analysis by FDA potentially would involve a deep analysis of the data comparing the risks to users and non-users (including children who may be future ENDS users) and the risks associated with youth addiction to nicotine against the data supporting the idea that ENDS represents a viable harm reduction opportunity. 

## 4. Conclusions

The first decade of federal regulation of ENDS has resulted in very limited controls placed on the market and a true epidemic of use of these products by youth. The FDA will soon determine how or whether these products should remain on the market. What had been a very small industry when the agency first sought to regulate ENDS has grown exponentially and is now largely controlled by the parent companies of cigarette manufacturers. These factors combine to create an especially challenging regulatory environment for FDA. Without the fast-tracked implementation of a strong federal regulatory response to the epidemic of youth vaping, we face the prospect of treating an entire generation of youth who are deeply addicted to nicotine.

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
