# Peer review of "Regulation of E-Cigarettes in the United States and Its Role in a Youth Epidemic"

_children, 2019, doi:10.3390/children6030040_

Round 1

Reviewer 1 Report

This paper provides a brief overview of the history of ENDS marketing in the US leading to the current ENDS epidemic among youth, and briefly describes regulatory efforts that have been suggested to tackle that epidemic.  This is a very important and timely issue, and well worth addressing, especially in a journal focusing on youth.  The author does a good job giving a broad, although brief, overview of the regulatory situation. Other than a few suggestions indicated in yellow comments in the attached PDF, my only concern about this paper is that it does not go into much detail about the regulatory efforts (or lack thereof), presumably because of word limits in the journal.  The paper would be even better if a few more sentences were added to the Conclusion that more fully describe the current regulatory situation, and dire implications of FDA's failure to act quickly.

Author Response

Please see "responses to reviewer 1.docx," which I have uploaded.

Reviewer 2 Report

This is a very well written, timely, and important review of the history of e-cigarette regulation in the US. I enjoyed reading it. I only have a few edits:

Line 14: Please delete the extra period at the very end

Line 85: Can you provide a little more detail (a sentence or two) about the nicotine salt and how it's different? I think that's a very important variable in the explanation of why Juul is dominating the market. 

Line 100: The sentence is missing a period

Line 110: Please add a comma after the word "minors"

Author Response

Thank you for your careful review.  Here are the changes made:

Comment 1: Line 14: Please delete the extra period at the very end

Response 1: The missing period has been added at line 14.

Comment 2 Line 85: Can you provide a little more detail (a sentence or two) about the nicotine salt and how it's different? I think that's a very important variable in the explanation of why Juul is dominating the market. 

Response 2: I have modified the end of the sentence (now at line 89) and included a sentence after to help explain the chemistry of Juul.  These sentences, beginning at line 89, now read: ". Its proprietary chemistry and development of “nicotine salts,” which deliver nicotine in a protonated rather than a free-base form, is certainly a part of the explanation [21]. As such, Juul delivers a different sort of nicotine than did its ENDS predecessors.  This is achieved through the use of benzoic acid to lower the pH of Juul’s liquid nicotine solution in order to aerosolize the drug in in its protonated form of highly palatable ultrafine particles [21]

Comment 3: Line 100: The sentence is missing a period

Response 3: I have found the missing period and restored it (now at line 107).

Comment 4: Line 110: Please add a comma after the word "minors"

Response 4: I have added the needed comma (now at line 117).